# Origins of chemoreceptor curvature sorting in *Escherichia coli*

Will Draper[1,2] & Jan Liphardt[1,2]

Bacterial chemoreceptors organize into large clusters at the cell poles. Despite a wealth of structural and biochemical information on the system's components, it is not clear how chemoreceptor clusters are reliably targeted to the cell pole. Here, we quantify the curvature-dependent localization of chemoreceptors in live cells by artificially deforming growing cells of *Escherichia coli* in curved agar microchambers, and find that chemoreceptor cluster localization is highly sensitive to membrane curvature. Through analysis of multiple mutants, we conclude that curvature sensitivity is intrinsic to chemoreceptor trimers-of-dimers, and results from conformational entropy within the trimer-of-dimers geometry. We use the principles of the conformational entropy model to engineer curvature sensitivity into a series of multi-component synthetic protein complexes. When expressed in *E. coli*, the synthetic complexes form large polar clusters, and a complex with inverted geometry avoids the cell poles. This demonstrates the successful rational design of both polar and anti-polar clustering, and provides a synthetic platform on which to build new systems.

[1] Biophysics Graduate Group and Department of Physics, University of California, Berkeley, California 94720, USA. [2] Bioengineering, Shriram Center for Bioengineering & Chemical Engineering, Stanford University, Stanford, California 94305, USA. Correspondence and requests for materials should be addressed to J.L. (email: jan.liphardt@stanford.edu).

A close examination of *Escherichia coli* reveals a highly spatially ordered environment, from a chromosome divided into hundreds of topologically isolated domains[1], to landmark proteins that oscillate from cell pole to cell pole to identify the cell's middle for division[2]. The components of other vital protein complexes, such as the phosphotransferase system[3] and the chemotaxis system[4], reside at the cell poles. Determining the molecular mechanisms that drive such spatial organization is crucial not only to understand bacterial physiology but also to engineering cells with novel functionalities[5].

The chemotaxis system is highly conserved and ubiquitous in bacteria, being found in a little over 50% of sequenced bacterial genomes[6]. The heart of the chemotaxis system is a large array of transmembrane chemoreceptors interconnected at their cytoplasmic distal tips by the histidine kinase CheA and a small adaptor protein CheW[4]. Chemoreceptors were first identified as a cluster at the cell pole in *E. coli* more than 20 years ago[7,8], and though chemoreceptor clusters have been found at the pole in all bacterial species studied thus far[9], little is known about how this crucial system is targeted there. Some bacteria contain a cryptic *parA* gene in the chemotaxis operon, which in the case of *Vibrio cholerae*[10] and cytoplasmic chemoreceptors in *Rhodobacter sphaeroides*[11] is necessary for polar targeting, but most bacteria including *E. coli* do not have such a *parA*. Previous research with *E. coli* has suggested that stochastic chemoreceptor cluster nucleation and growth could propagate an existing polar localization pattern to daughter cells, either directly[12], or in conjunction with unspecified cytoskeletal elements[13,14]. Other researchers have found that polar cluster localization in *E. coli* requires an intact Tol/Pal system[15].

One simple mechanism that could drive chemoreceptor clusters to the cell poles is the higher mean curvature of the membrane at the cell pole compared with the rest of the cell. Although chemoreceptors do not have any obvious structural similarity to well-known membrane curvature sensing and generating domains[16], electron microscopy has shown that overexpressed *E. coli* serine chemoreceptor, Tsr, forms tubular membrane invaginations and apparent vesicular structures[17]. Subsequent cryo-electron tomography[18,19] and cross-linking[20] have found that chemoreceptors form a cone-shaped trimer-of-dimers (ToD) *in vivo*, corroborating a previous crystal structure of the cytoplasmic domain of Tsr[21]. This ToD geometry led Endres[22] to propose a curvature-based mechanism for polar localization of the array. Recent work in *Bacillus subtilis* supported this notion, concluding that TlpA, which is homologous to *E. coli* chemoreceptors, is sorted by membrane curvature[23].

Here we directly quantify the curvature-dependent localization of *E. coli* chemoreceptors *in vivo* by artificially deforming growing cells in curved agar microchambers, as inspired by Takeuchi *et al.*[24]. We find that *E. coli* chemoreceptor clusters are highly enriched in curved membranes, and that the magnitude of this curvature sensitivity can fully explain their polar localization. The scaling of the enrichment as a function both of cluster size and membrane curvature matches predictions from a recent model for chemoreceptor clustering[25]. We propose that chemoreceptor ToDs adopt a curved cone shape to maximize conformational entropy, presenting a novel mechanism for curvature sensitivity. We use this framework to design a series of synthetic curvature-sensitive protein complexes that sort into regions of either positive or negative curvature in living *E. coli*, closely mimicking the localization pattern of natural chemoreceptor clusters.

## Results

**Artificially curved cells**. To test the possibility that *E. coli* chemoreceptor clusters localize to cell poles in response to membrane curvature, we artificially deformed growing filamentous *E. coli* in agar microchambers (Supplementary Fig. 1) using a technique developed by Takeuchi *et al.*[24]. Cells deformed into a variety of irregularly curved shapes as they elongated into long filaments and contacted the edges of the agar chambers (Fig. 1a), providing a range of local curvatures along the cell's length. We fluorescently labelled chemoreceptor clusters using a fusion of eYFP to the methyltransferase CheR, which binds to the C terminus of both high-abundance *E. coli* chemoreceptors (Tsr and Tar) and is not thought to impair clustering when fused to eYFP[13] (Fig. 1b).

Cell outlines were segmented from brightfield images using a sub-pixel contour-tracing algorithm (Fig. 1c). We fit a third-order polynomial to a sliding window along the cell outline to measure the first and second derivatives of position with respect to distance, from which we could directly calculate local curvature (Fig. 1d, Supplementary Fig. 2). We define positive curvature as cell-pole-like curvature, for example, the inside face of a convex shape. Cell poles were identified in each outline as the two regions with the highest positive curvature and were ignored for all further analyses. Profiles of lateral cell curvature clearly reflected the curvature imposed by the agar chambers (Fig. 1c,d), typically on the order of $\pm 0.5\,\mu m^{-1}$. We observed an additional irregular variation on the order of $\pm 0.25\,\mu m^{-1}$ occurring approximately every $\sim 1\,\mu m$, in close accordance with other published work[26]. These two sources of variation together gave a broad distribution of measured lateral cell curvatures, with 95% of measured curvatures between $-0.73$ and $0.70\,\mu m^{-1}$ (Fig. 2b).

Individual chemoreceptor clusters were segmented from fluorescence images and localized using a two-dimensional Gaussian fit (Fig. 1c). Each cluster within $0.3\,\mu m$ of the cell outline was assigned the curvature from the closest point along the cell outline, and as a measure of relative cluster size, each cluster was also assigned the integrated, background-subtracted fluorescence intensity (Fig. 1c,d). The resultant distribution of cluster intensities was roughly exponential, in close accordance with previously measured cluster size distributions[12] (Fig. 2a).

**Chemoreceptor distributions in curved cells**. We plotted a histogram of cluster frequency as a function of lateral membrane curvature, weighted to correct for non-uniform sampling of cell curvature, and normalized to one at zero curvature. We found a large enrichment of clusters with increasing lateral membrane curvature, appearing approximately exponential (Fig. 2c), demonstrating that chemoreceptor localization is sensitive to membrane curvature. We separated clusters into low-, medium- and high-intensity groups as a proxy for cluster size (Fig. 2a), and for each group, we measured the curvature enrichment. Larger clusters were markedly more enriched in high curvature regions than small clusters (Fig. 2c).

To examine the effect of cluster size more systematically, we binned the raw cluster data into 100 sequential, overlapping intensity bins, and fit a simple exponential $F = e^{E \times C}$ to each bin's curvature enrichment profile, where $F$ is the cluster frequency, $C$ is the curvature and the sole fit parameter $E$ is the enrichment magnitude. Enrichment magnitude $E$ describes the relative frequency of a cluster per unit curvature compared to un-curved lateral cell. We found that $E$ scales roughly linearly as a function of cluster size (Fig. 2d), allowing us to extrapolate the enrichment magnitude for a large polar cluster ($E_{polar} = 6.7\,\mu m$, Fig. 2d inset). Using $E_{polar}$ and a nominal polar curvature of $2.5\,\mu m^{-1}$, we estimate that an average polar cluster will be $F = e^{6.7 \times 2.5} = 0.5 \times 10^8$ more likely to be at the cell pole than in the lateral membrane, more than enough enrichment to fully explain why large receptor clusters are always at the cell pole.

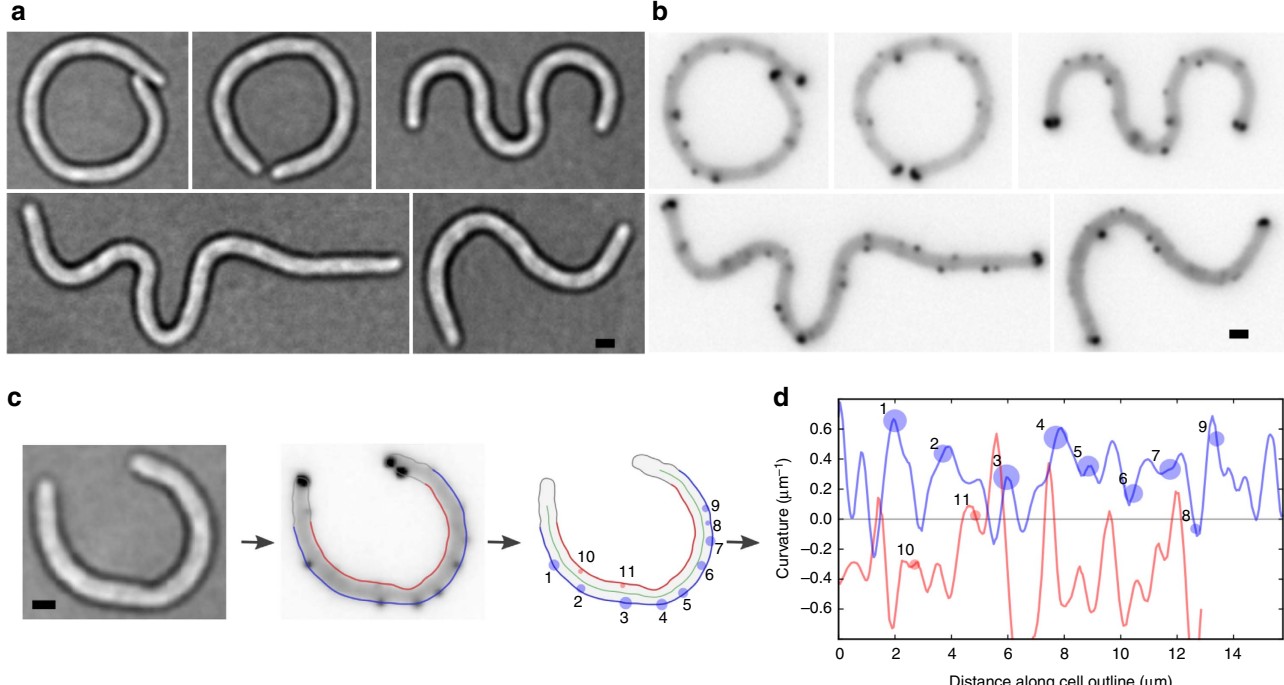

**Figure 1 | Filamentous *E. coli* artificially curved in agar microchambers** (**a**) *RP437 E. coli* were grown in curved agar microchambers cast in M9 minimal media containing 0.4% glycerol and 50 μg ml$^{-1}$ cephalexin to induce filamentation. (**b**) Native chemoreceptor clusters were labelled by expressing YFP-CheR, and imaged using epi-fluorescence. (**c**) Cell outlines were traced from brightfield images using a sub-pixel contour algorithm. Corresponding chemoreceptor clusters were segmented from fluorescence images, and assigned a position along the cell outline (numbered circles, size indicates intensity). (**d**) Curvature along the cell outline was measured, and each cluster was assigned a curvature based on its position along the outline (numbered circles). Scale bars, 1 μm.

Next, we wanted to determine whether curvature enrichment depends on any external factors, or is an intrinsic property of chemoreceptors. Other researchers have suggested that localization depends on unspecified cytoskeletal components[13,14]. To test this possibility, we measured the curvature enrichment profile of receptor clusters in an *RP437* mutant with a complete knockout of the Min system (*RP437 ΔminCDE*) (Fig. 3a, Supplementary Fig. 4). The Min system helps direct division site placement by oscillating from cell pole to cell pole[27]. We also tested whether the anionic lipid cardiolipin, which is found in micro-domains at the cell pole and is enriched in regions of positive curvature[28], has a role in targeting chemoreceptors to the pole by measuring the curvature enrichment profile of chemoreceptor clusters in an *RP437* mutant knocked out for all three cardiolipin synthases—ClsA, ClsB and ClsC (*RP437 ΔclsABC*). In both cases, we found that curvature enrichment was indistinguishable from receptor curvature enrichment in wild-type cells (Fig. 3a, Supplementary Fig. 4).

We also measured curvature enrichment profile of receptor clusters in *RP437* mutants knocked out for CheA and CheW (*RP437 ΔcheAW*), which along with chemoreceptors form the structural core of chemoreceptor clusters. Previous imaging in *ΔcheAW* strains has found chemoreceptors as diffuse polar caps[8] that are undetectable in cryo-electron tomography[18]. However, we found chemoreceptor clusters as distinct foci along the lateral cell, in addition to the previously described polar caps (Fig. 3b, Supplementary Fig. 4). Though clusters in *ΔcheAW* cells were on average much smaller than native clusters (Fig. 3b inset), we found that the curvature enrichment profile was indistinguishable from wild-type chemoreceptor clusters when controlling for cluster size (Fig. 3b).

Finally, we tested whether defects in the Tol/Pal system affect curvature enrichment, as recent research has found that an intact Tol/Pal system is required for chemoreceptors to localize to the poles[15]. *RP437* mutants knocked out for either Tol or Pal (*RP437 Δtol, RP437 Δpal*) showed phenotypic defects when grown in suspension, in accordance with the Tol/Pal system's vital role in cell physiology, and quickly lysed when we attempted to grow them as filaments in our agar chambers. However, we were able to grow artificially deformed *RP437* mutants knocked out for Lpp (*RP437 Δlpp*) (Supplementary Fig. 4), a peripheral component of the Tol/Pal system that has also been reported to affect polar targeting. We found that chemoreceptors are curvature sensitive in *Δlpp* cells, but much like in *ΔcheAW* cells, the average cluster size is smaller (Fig. 3b). When controlling for size, the receptor curvature enrichment profile in *Δlpp* cells was indistinguishable from the enrichment profile in wild-type *RP437* cells (Fig. 3b). Chemoreceptor clusters in non-filamentous, suspension-grown *Δtol* cells were on average smaller than clusters in *Δlpp* cells, and markedly smaller than clusters in wild-type RP437 cells (Fig. 3c). Thus, given the strong effect of cluster size on enrichment magnitude, these results suggest that defects in the Tol/Pal system affect polar clustering primarily by reducing average cluster size, but does not affect cluster curvature sensitivity.

**Receptor intrinsic curvature.** These observations indicate that curvature sensitivity is an intrinsic property of chemoreceptors, and does not depend on any external factor. Previous work by Endres[22] and Haselwandter and Wingreen[25] have suggested that the curved cone shape of a receptor ToD is sufficient to explain curvature sorting of clusters. In both models, the cone shape induces an elastic deformation in the lipid bilayer, incurring an energy penalty that scales roughly linearly with the mean curvature of the surrounding membrane (Fig. 4a). The difference in deformation energies $\Delta E$ between a ToD in a curved versus a flat membrane dictates the probability $F$ of

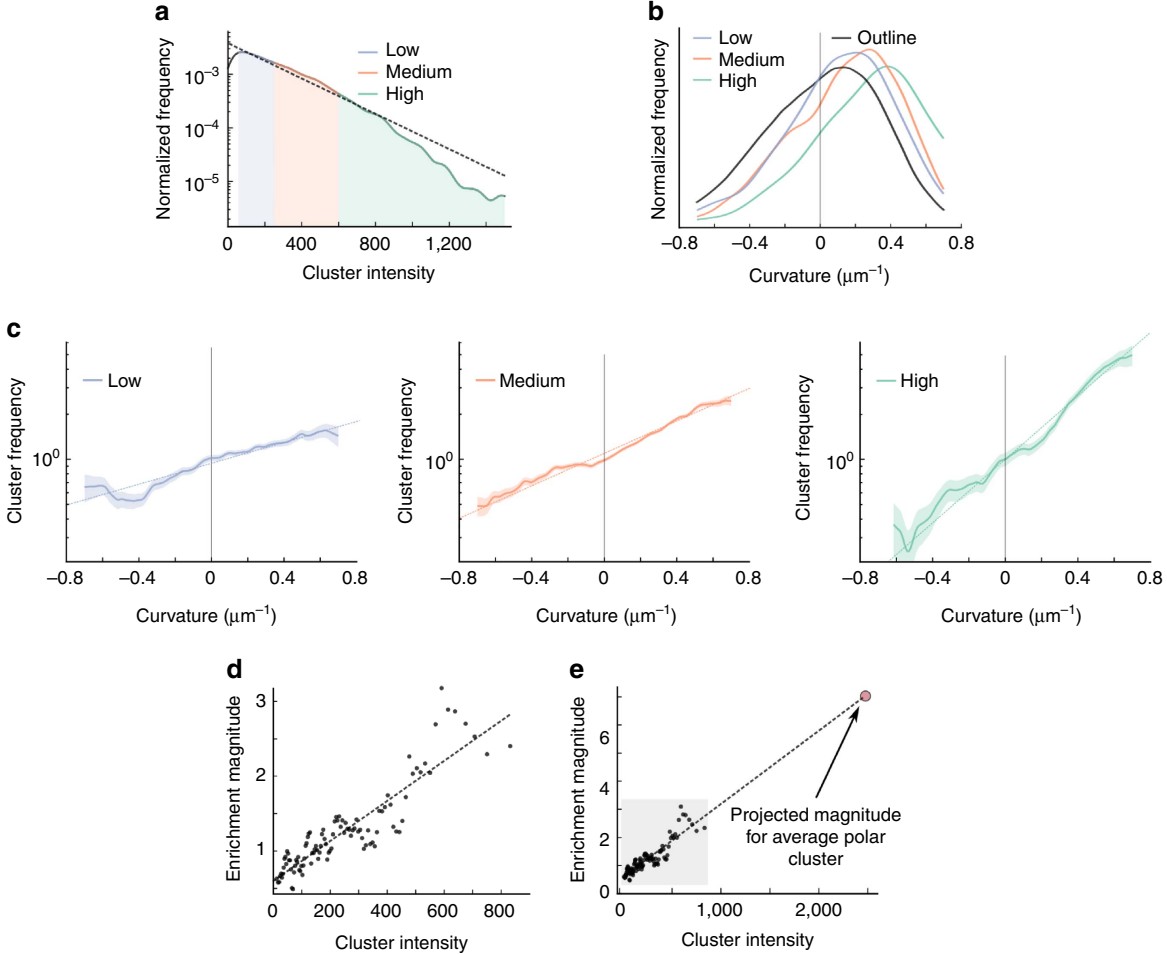

**Figure 2 | Enrichment of chemoreceptor clusters with curvature in wild-type *RP437 E. coli* (a)** Histogram of chemoreceptor cluster intensities, showing a roughly exponential distribution of intensities (dotted line). Clusters were split into low-, medium- and high-intensity bins for further analysis. (**b**) Histogram of curvatures measured for all cell outlines, compared to curvatures assigned to low-, medium- and high-intensity clusters. (**c**) Histogram of cluster frequency as a function of curvature, normalized to 1 at zero curvature. Histograms were weighted by the whole-data set curvature distribution to correct for uneven sampling of curvatures. Dotted lines represent a single exponential fit to the data, and shaded areas indicate $\pm 1$ s.d. (**d**) Enrichment magnitude as a function of cluster intensity, measured by fitting cluster data from multiple overlapping intensity bins to a single exponential $F = e^{E \times C}$, where $F$ is the frequency, $C$ is the curvature and $E$ is the enrichment magnitude, the slope of the fits in **c**. Enrichment magnitude increased approximately linearly with cluster intensity (dotted line). (**e**) Extrapolation of the enrichment magnitude for the intensity of a large polar cluster, shaded area corresponds to the region shown in **d**. Data were collected from 649 cells totalling 24.9 mm of cell outline, and 5,181 clusters.

finding a ToD in the curved membrane according to the Boltzmann distribution $F = e^{-\Delta E / kT}$, where $T$ is the temperature and $k$ is the Boltzmann constant. Additionally, in both models, coupling between ToDs leads the energy difference to scale roughly linearly with the size of a cluster. Thus, energy $\Delta E$ should scale approximately linearly with both cluster size and membrane curvature, with the resulting cluster frequency $F$ in turn varying exponentially. Both these predictions are in good agreement with our measurements.

Next, we sought to investigate the source of the curved ToD conformation. Most chemoreceptors have a large, periplasmic ligand-binding domain, which could conceivably force the trimer of dimers structure open into a cone shape, with the trimerization domain acting as a hinge (Fig. 4a). However, the curvature enrichment profile of Tar° clusters, a signalling-competent mutant of *E. coli* Tar that lacks the periplasmic domain[29], expressed alone in otherwise receptorless *UU2611* cells was indistinguishable from the receptor enrichment profile of wild-type receptor clusters (Fig. 4b, Supplementary Fig. 4). This suggests that the periplasmic domain has no role in the cone-like ToD shape.

It is also possible that protein–protein contacts in the trimerization domain hold the cone shape open, effectively cantilevering the transmembrane domains apart. Recent work in *B. subtilis* with the chemoreceptor TlpA, which also forms ToDs, suggested that curvature sensitivity arises from rigidity within the ToD cone shape holding the transmembrane domains apart and at an angle in the membrane[23]. The researchers reported that curvature sensitivity was eliminated by the introduction of two glycine residues within the receptor HAMP domain, increasing receptor flexibility. To test this in *E. coli*, we made homologous glycine mutations in the HAMP domain of Tsr, making Tsr-flexiHAMP, and expressed it in otherwise receptorless *UU2611*. Although clustering was severely impaired, appearing nearly homogenous throughout the membrane, there were frequent small clusters (Fig. 4b, Supplementary Fig. 4). These small Tsr-flexiHAMP clusters showed similar curvature sensitivity to wild-type chemoreceptors, when controlling for cluster size (Fig. 4b). Thus, just like cells with an impaired Tol/Pal system, the Tsr-flexiHAMP mutant affects polar localization by reducing average chemoreceptor cluster size, but has no effect on chemoreceptor curvature sensitivity.

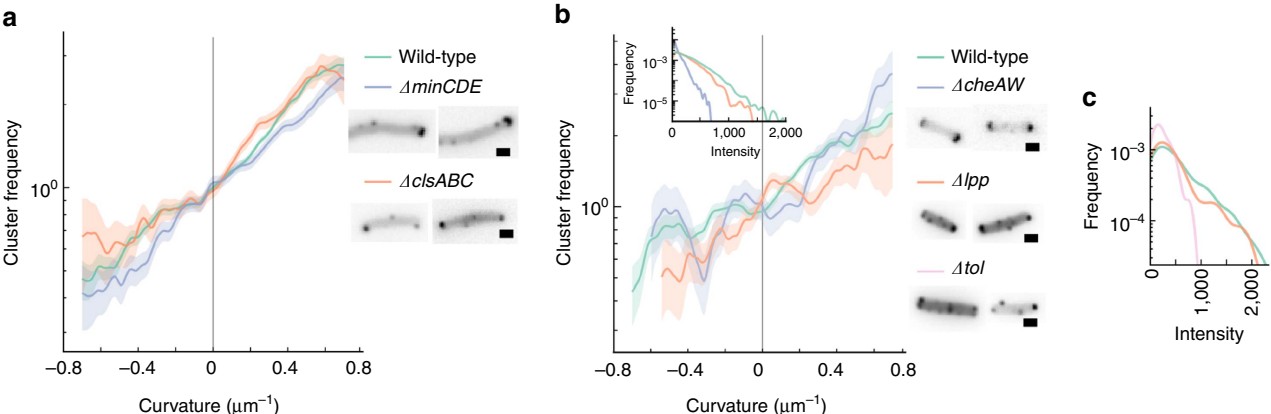

**Figure 3 | Chemoreceptor cluster frequency as a function of curvature in various mutant backgrounds. (a)** Chemoreceptor cluster frequency as a function of curvature in *RP437* labelled with YFP-CheR (*N* = 649 cells, 24.9 mm outline, 3,828 clusters), compared to Min knockout mutants (*RP437 ΔminCDE*, *N* = 218 cells, 7.1 mm outline, 899 clusters), and mutant cells knocked out for all three cardiolipin synthases (*RP437 ΔclsABC*, *N* = 219 cells, 8.7 mm outline, 1,272 clusters), with sample micrographs of each mutant. **(b)** Chemoreceptor cluster frequency as a function of curvature in *RP437* labelled with YFP-CheR (*N* = 649 cells, 24.9 mm outline, 1,305 clusters), compared to Lpp knockout mutants (*RP437 Δlpp*, *N* = 171 cells, 8.8 mm outline, 191 clusters), and cells knocked out for CheA and CheW (*RP437 ΔcheAW*, *N* = 96 cells, 4.6 mm outline, 285 clusters). Cluster frequency was only measured from low-intensity clusters, as *ΔcheAW* cells had few large clusters. Inset shows corresponding receptor cluster size distributions. Sample micrographs are provided for each mutant, and a Tol knockout (*RP437 Δtol*). **(c)** Receptor cluster size distribution from suspension grown, non-filamentous wild-type *RP437* cells (*N* = 62 cells, 195 clusters), compared to Lpp knockouts (*RP437 Δlpp*, *N* = 47 cells, 139 clusters), or Tol knockouts (*RP437 Δtol*, *N* = 22 cells, 47 clusters), with colouring corresponding to **b**. Shaded areas indicate ±1 s.d. Scale bars, 1 μm, and all sample micrographs are from suspension grown, non-filamentous cells.

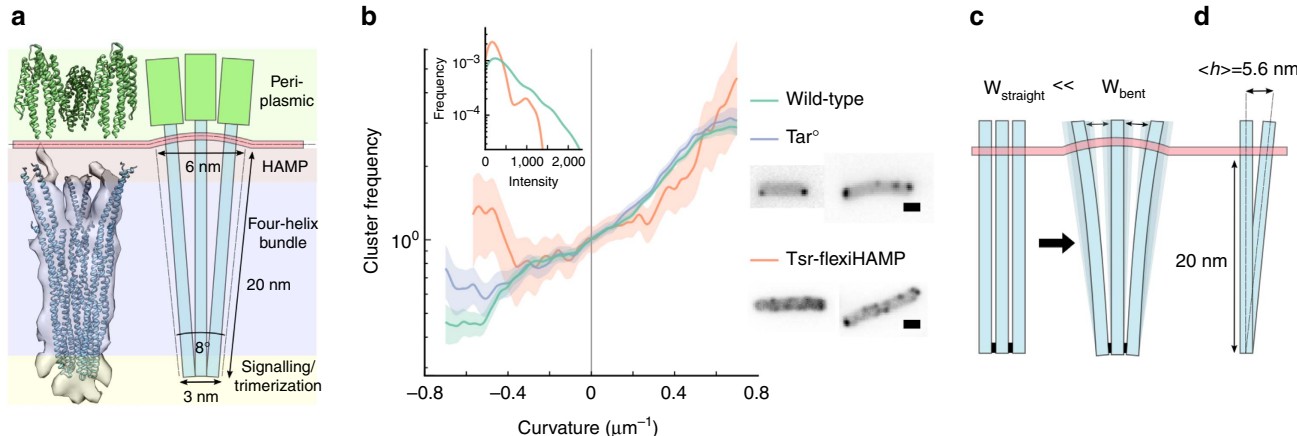

**Figure 4 | Intrinsically curved shape of chemoreceptor trimer of dimers. (a)** Left shows crystal structures of the Tsr cytoplasmic domain (1QU7) and periplasmic domain (3ATP) superimposed on a cryo-electron tomographic reconstruction of native chemoreceptor trimers of dimers (EMDB-2158). Right shows a cartoon representation of the trimer of dimers structure, with different subdomains highlighted and corresponding dimensions. **(b)** Chemoreceptor frequency as a function of curvature for wild-type *RP437* labelled with YFP-CheR (*N* = 649 cells, 24.9 mm outline, 3,105 clusters), compared to receptorless *UU2611* cells that express a mutant of Tar that lacks the periplasmic domain (Tar°, *N* = 197 cells, 8.7 mm outline, 1,088 clusters) as the only chemoreceptor, and receptorless *UU2611* cells that express a mutant of Tsr with a glycine hinge added to the HAMP domain (Tsr-flexiHAMP, *N* = 108 cells, 3.9 mm outline, 150 clusters) as the only chemoreceptor. Shaded areas indicate ±1 s.d. **(c)** Schematic depicting the difference in conformational entropy for three transmembrane flexible rods tethered together at one end. On the left is a single conformation where all three rods pass perpendicularly through the membrane ($W_{straight}$), and on the right a family of conformations where the rods are bent and pass through the membrane at an angle ($W_{bent}$). **(d)** Schematic showing the average deflection $<h>$ for a four-helical bundle with a persistence length of 500 nm. Shaded areas indicate ±1 s.d. Scale bars, 1 μm.

A final possibility is that the transmembrane domains of a ToD spread apart to maximize conformational entropy, while the cytoplasmic-distal ends are held tightly in together by protein–protein contacts along the trimerization domain (Fig. 4c). On the length scale of a chemoreceptor, protein is relatively flexible[30]. Using measurements from α-helical coiled-coils[31], a conservative estimate for 4-helix bundle persistence length is $P = 500$ nm. A single receptor dimer with a length of $L = 20$ nm should experience an average deflection along its length of $\langle h \rangle = (L^3/P)^{1/2} = 4.0$ nm due to thermal motion within the four-helix bundle alone (Fig. 4d), more than enough to account for the cone shape we estimate from structural data where the average deflection is 1.4 nm from perpendicular. Additional flexibility could come from a ring of conserved glycine residues within four-helix bundle[32]. Thus, ToD entropy is maximized in an intrinsically curved cone configuration because there are many possible conformations where the transmembrane domains are splayed apart due to flex in the four-helix bundles, but only one

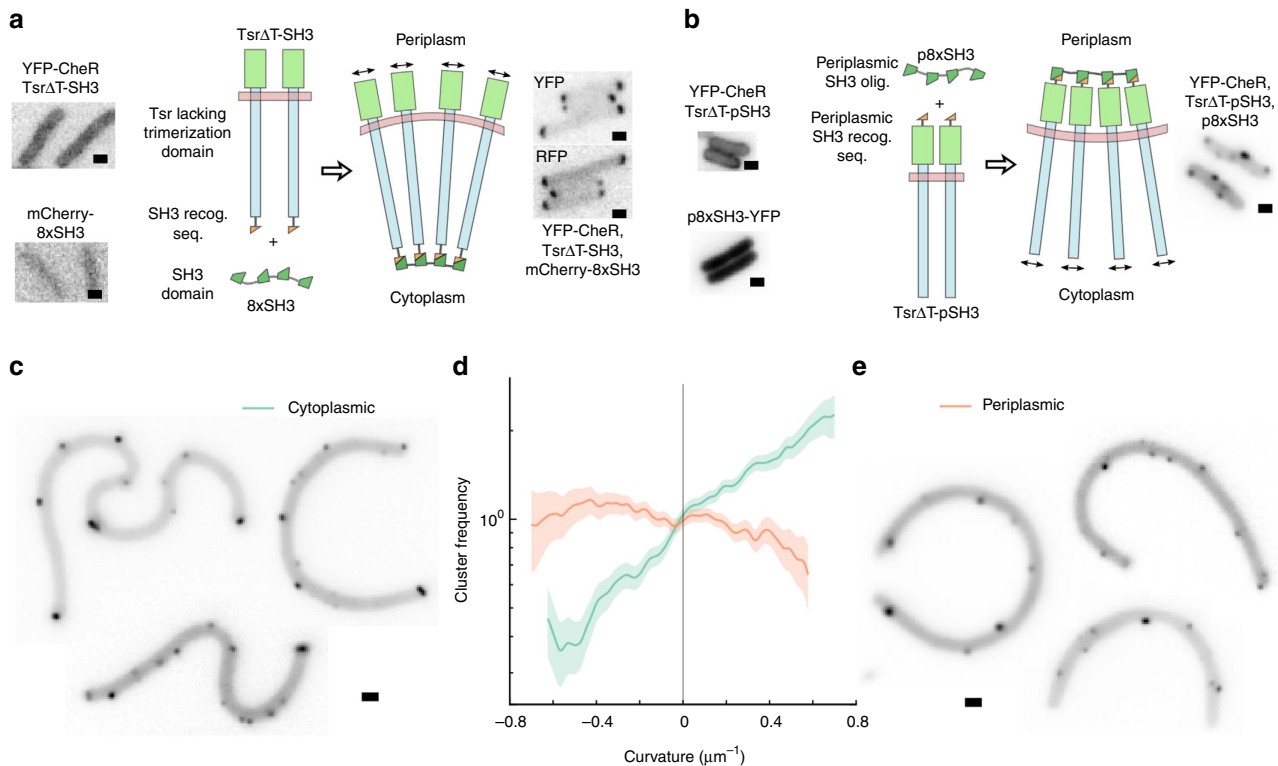

**Figure 5 | Artificial curvature-sensitive complexes using protein scaffolds.** (**a**) Artificial curvature-sensitive protein complexes were built out of long rod proteins, in this case a mutant of Tsr with the cytoplasmic trimerization domain entirely replaced by an Src homology 3 (SH3) recognition sequence, and a multivalent protein scaffold that binds to the rod proteins, composed of eight direct repeats of the cognate SH3 domain. By tethering the rods together by only one end, the design forces the other end to spread apart to maximize conformational entropy, tilting the transmembrane domains. The components were imaged using YFP-CheR as a label for receptors, and a direct fusion of mCherry to the scaffold protein. The components were expressed alone (left side), and together (right side). (**b**) Artificial protein complexes were built with a different connectivity to flip curvature sensitivity. Complexes used identical components, except the SH3 recognition sequence on Tsr was moved to the periplasmic domain, and the scaffold protein was exported to the periplasm using the twin arginine transporter (TAT) system. Receptors were labelled with YFP-CheR, and were expressed either alone (left side), or together with periplasmic scaffold (right side). (**c**) Cells expressing cytoplasmic scaffolds from **a** were artificially deformed in agar microchambers ($N = 375$ cells, 15.0 mm outline, 857 clusters) to measure (**d**) cluster frequency as a function of curvature in comparison to (**e**) artificially deformed cells expressing periplasmic scaffolds ($N = 394$ cells, 17.0 mm outline, 778 clusters) from **b**. Shaded areas indicate ±1 s.d. Scale bars, 1 μm.

where the transmembrane domains all pass perpendicularly through the membrane.

**Synthetic clusters using scaffolds**. Next, we attempted to re-engineer the splayed, conical ToD shape into a synthetic protein complex to test whether our conformational entropy model can explain the origins of curvature sensitivity. We used long, transmembrane rod proteins to mimic chemoreceptors, and a separate, multivalent scaffold protein that binds to the cytoplasmic distal tip of the rod proteins to form a large oligomer. With the cytoplasmic tips held together, the transmembrane domains should spread apart and tilt to maximize conformational entropy, generating small deformations in the membrane that are minimized in membranes with inward curving, positive curvature (Fig. 5a).

We built a scaffold protein using eight direct repeats of the Src homology 3 (SH3) domain from mouse CRK, which binds to a short, engineered SH3 recognition sequence with high affinity ($K_d = 0.1\,\mu M$)[33]. Each SH3 repeat was separated by a flexible five-amino acid glycine–serine linker, making 8xSH3 (Fig. 5a, Supplementary Fig. 5). As a rod protein, we designed a mutant of the chemoreceptor Tsr with the entire trimerization/signalling domain deleted, as identified by a bioinformatic comparison of all known chemoreceptor sequences[34], encompassing 25 amino

acids on either side of the central tip residue E391. The trimerization/signalling domain was replaced with the 11-amino acid SH3 recognition sequence, flanked by two flexible 5-amino acid linkers, making TsrΔT-SH3 (Fig. 5a, Supplementary Fig. 5). The two components were expressed in receptorless *UU2611* cells to eliminate any interference from native chemoreceptor clusters. When expressed alone, TsrΔT-SH3 labelled with YFP-CheR appeared diffuse throughout the membrane, with no apparent clustering or aggregation, suggesting that the protein can freely diffuse within the membrane and is properly folded. Similarly, a direct fusion of mCherry to 8xSH3 expressed alone appeared diffuse throughout the cytoplasm (Fig. 5a).

When both rod and scaffold proteins were expressed together, our synthetic complex formed efficiently, with the components co-localized as punctate spots (Fig. 5a). The complexes were predominantly at the cell pole, with smaller clusters along the lateral membrane in a manner indistinguishable from wild-type chemoreceptor clusters. The curvature enrichment profile of the artificial complex showed a strong preference for positive curvature, similar to the curvature enrichment of native chemoreceptor clusters (Fig. 5c,d). Both clustering and polar localization were robust to variation in component expression level. Whole-cell cryo-electron tomograms showed no evidence of protein aggregation or membrane distortion, even with both components grossly overexpressed, and also showed no evidence

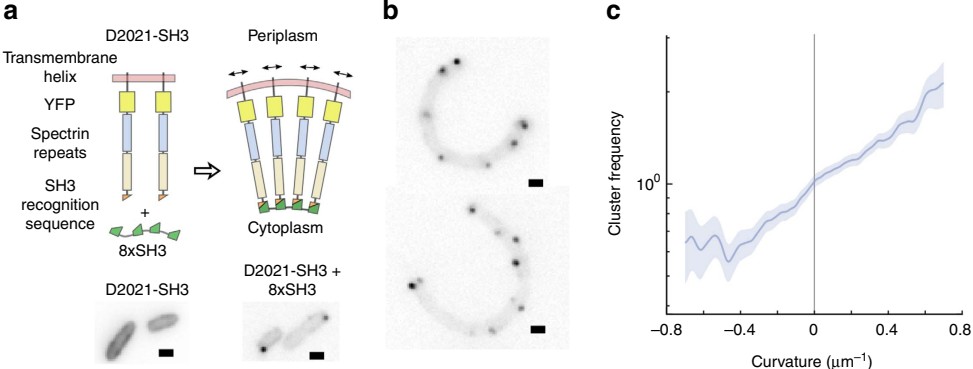

**Figure 6 | Artificial curvature-sensitive protein complex engineered from heterologous components.** (**a**) Complexes were built from multivalent scaffold proteins consisting of eight direct repeats of an SH3 domain, and rod proteins consisting of a cyanobacterial transmembrane helix, YFP, two monomeric spectrin repeats from human dystrophin and an SH3 recognition sequence. Rods imaged directly via the internal YFP, either alone (left side), or with the scaffold protein (right side). (**b**) Cells expressing heterologous rods and scaffolds were artificially deformed in agar microchambers to measure (**c**) cluster frequency as a function of curvature ($N = 375$ cells, 15.0 mm outline, 857 clusters). Shaded areas indicate $\pm 1$ s.d. Scale bars, 1 μm.

of an ordered transmembrane array like would be observed for a native chemoreceptor cluster (Supplementary Fig. 3).

We hypothesized that we could invert the curvature sensitivity of synthetic clusters to prefer negative curvature by connecting the rod proteins together by their periplasmic domains, analogous to inverting the orientation of the trimer of dimers in the membrane so the cone vertex is in the periplasm (Fig. 5b). To test this, we designed a Tsr mutant, TsrΔT-pSH3, with the cytoplasmic trimerization domain replaced by a flexible 5-amino acid linker, and the 11-amino acid SH3 recognition sequence inserted into an exposed loop of the periplasmic domain, again flanked by two flexible 5-amino acid linkers (Supplementary Fig. 5). We targeted the scaffold protein to the periplasm using the twin arginine transporter (TAT) signal sequence from *E. coli* TorA (ref. 35) (Supplementary Fig. 5).

Periplasmically scaffolded complexes readily formed when the components were co-expressed in cells, appearing as punctate spots on the cell membrane (Fig. 5b). As before, when either component was expressed in isolation, we saw no clustering. The periplasmic complexes appeared randomly distributed across the cell membrane, with the largest clusters almost never at the cell pole (Fig. 5b,e). In accordance with our conformational entropy model, the curvature enrichment profile of the periplasmic complex revealed a modest preference for negative curvature, opposite to the curvature of the cell pole (Fig. 5d,e). Thus, by merely changing the connectivity of the complex, we were able to markedly alter localization, suggesting that the complex's shape is likely the sole factor responsible for curvature sensitivity.

We expect that the periplasmic complex is less curvature sensitive than the cytoplasmic complex is because in the periplasmic complex, the SH3-recognition sequence is several-fold closer to the cell membrane. The entropic spreading model predicts that the SH3-recognition sequence is effectively a hinge point in the diverging ToD like conformation, with the divergence driven by flexibility along the rod protein's length. Thus, when the hinge point is closer to the membrane, there is less deflection at the membrane, and less curvature enrichment. A second key difference is that the periplasmic complex induces a curvature sign mismatch. The periplasmic complex has negative curvature with respect to a membrane that is positively curved, whereas the cytoplasmic complex has positive curvature with respect to a less positively curved membrane. The energetics of such a sign mismatch are likely much more complicated than the energetics of the cytoplasmic complex, in ways that current models do not attempt to address.

**Totally synthetic clusters.** To control for any contribution from TsrΔT-SH3 to our artificial complex's curvature sensitivity unrelated to its structural role as a rod, we designed an entirely heterologous membrane anchored rod. As a membrane tether, we used a 22-amino acid single-pass transmembrane helix from an arbitrary *Synechocystis* gene, which we fused via a 7-amino acid glycine–serine linker to the N terminus of YFP. We built an extended rod domain by fusing spectrin repeat domains from human dystrophin, which form long structural filaments, to the C terminus of YFP. We used dystrophin spectrin repeats 20 and 21, together extending ∼10 nm, as they have been previously shown *in vitro* to have no propensity to self-aggregate or oligo-merize, and no affinity for membranes[36,37]. Finally, we added an SH3 binding sequence to the C terminus of the spectrin repeat domains, to make D2021-SH3 (Fig. 6a, Supplementary Fig. 5).

The heterologous rod distributed evenly across the membrane (Fig. 6a) when expressed alone in *E. coli* DH5α, identically to our previous rod, TsrΔT-SH3. When co-expressed with our cyto-plasmic scaffold 8xSH3, the heterologous rod efficiently formed clusters, which were predominantly localized to the poles (Fig. 6a,b). The curvature enrichment profile of the entirely heterologous complex was nearly identical to the profile of our receptor-based artificial complex, with a strong preference for positive curvature (Fig. 6c).

### Discussion

In summary, we have found that *E. coli* chemoreceptor localization is highly sensitive to membrane curvature. The sensitivity we measured is strong enough to fully explain the localization of large receptor clusters at the cell poles, and appears to be an intrinsic property of chemoreceptors. Finally, we refactored the structure and connectivity of a chemoreceptor cluster using entirely heterologous components to make a synthetic, curvature-sensitive protein complex.

We propose that chemoreceptor curvature sensitivity is a result of conformational entropy within chemoreceptor trimer of dimers spreading the transmembrane regions apart, leading to the cone shape seen in structural studies. We used the entropic spreading framework to impart curvature sensitivity onto a series of synthetic protein complexes built from uncurved components. The complexes consisted of long, transmembrane rods connected together by either their cytoplasmic or periplasmic domains, leading to complexes that prefer positive or negative membrane curvature, respectively.

With chemoreceptor clusters and more generally our entropic spreading model, we have demonstrated a unique mechanism for membrane curvature sensing. Artificial confinement of proteins bound to the outside of large vesicles within small lipid domains was recently shown to induce extensive membrane tubulation[38]. This curvature-generating effect was modelled as an imbalance in lateral steric pressure across the two bilayer leaflets, caused by a high density of bulky protein on one side of the membrane, but no protein on the other side. Recent work has found that mammalian Epsin1 and AP180, both involved in clathrin-based vesiculation, induce membrane curvature using such a steric mechanism[39]. Our synthetic complexes and by extension trimers-of-dimers employ a similar principle, except our synthetic complexes experience an imbalanced compressive force, driven by binding to the SH3 scaffold protein. This attraction-driven compressive force, in contrast to a repulsive steric force, reverses the direction of the induced curvature, instead bringing the membrane toward the protein. Such a compressive mechanism has been predicted in the context of self-attractive membrane adsorbed polymers[40] and protein clusters[41], and has been theorized to play a role in the sorting of some GPI-anchored proteins and regulated secretory proteins (RSPs) in the highly curved *trans*-Golgi network[41].

## Methods

**Strains and plasmids.** Except where noted, all cell strains were derivatives of the *E. coli* K-12 strain RP437, including receptorless strain UU2611 (*tsr tar tap trg aer cheR*). Both strains were a gift from Sandy Parkinson (University of Utah, Salt Lake City). Knockout strain RP437 Δ*clsABC* was made by sequential P1 transductions from Keio clones[42] JW1241-5 (Δ*clsA*), JW0772-1 (Δ*clsB*) and JW5150-1 (Δ*clsC*) (Yale Stock Center). Keio clones are kanamycin-marked knockouts of complete open reading frames (ORFs) in K-12-derived *E. coli*. Knockouts for Tol and Lpp were made by P1 transduction from Keio clones JW0729-3 (Δ*tol*) and JW1667-5 (Δ*lpp*) (Yale Stock Center). Knockout strains RP437 Δ*minCDE* and RP437 Δ*cheAW* were made by P1 transduction from donor strains made directly through lambda red recombination, according to Datsenko and Wanner[43]. In both cases, all ORFs were removed in their entirety, from the beginning of *minC* to the end of *minE* (which includes *minD*), and from the beginning of *cheA* to the end of *cheW*. Unless otherwise noted, cells were propagated in standard LB medium at 37 °C. All base plasmids were a gift from Sandy Parkinson (University of Utah, Salt Lake City).

All plasmids, except for the scaffold 8xSH3, were made directly using sequence- and ligase-independent cloning (SLIC)[44], a PCR-based recombination method. As PCR is unreliable for amplifying direct repeats, the scaffold 8xSH3 was made using a restriction enzyme-based BglBrick-like assembly[45], and propagated in Stbl2 (Invitrogen). BglBrick assemblies make it possible to iteratively double the number of repeats in a construct through subsequent rounds of cloning, taking advantage of compatible restriction sites that can anneal to leave behind an uncleavable 6-nucleotide scar. All chemoreceptor mutants, chemoreceptor derivatives and transmembrane rods were expressed off derivatives of pRR48 (ref. 46). pRR48 is pBR322 derivative with an IPTG-inducible *tac* promoter, which unless otherwise noted was induced with 100 µM IPTG. Scaffold proteins were expressed off a derivative of pKG116, a salicylate-inducible derivative of pACYC184. Unless otherwise noted, all scaffold proteins were induced with 0.4 µM salicylate. YFP-CheR was expressed off of a plasmid derived from the arabinose-inducible pVS102 (ref. 8), and induced with 0.01% arabinose. All plasmid and protein sequences are available at the Harvard Dataverse.[47]

**Microchambers.** Silicon wafers (4″) were coated with FujiFilm OCG-825 G-line resist on a SVG 8626, exposed to a 10″ printed chromium mask (Fine Line Imaging) using a 10 × GCA 6200 wafer stepper, and developed on a separate SVG 8626. Agar microchambers were made by casting low melt agar (3% w/v) containing M9 minimal media and the division inhibitor cephalexin (50 µg ml$^{-1}$) directly on the developed photoresist. A suspension of exponential cells growing in M9 with 0.4% glycerol were sandwiched between the agar chamber and a microscope coverslip, where there were grown at 33 °C for 3–4 h before imaging.

**Microscopy.** All microscopy was performed using a 1.4 NA × 100 objective on a Nikon TE2000 inverted microscope, equipped with a Sutter Instrument Lambda LS xenon arc lamp and an Andor iXon+ EMCCD. We used the Nikon filter set G2E/C for imaging RFP and FM-4-64, and custom filters for imaging YFP (505/20 excitation, 520LP emission). All image acquisition and microscope control was performed using custom software (C++, Python).

**Data processing.** Images were analysed using custom written software written entirely in C++ and Python, drawing heavily on the NumPy, SciPy and Matplotlib Python modules. Cell outline segmentation was performed using a sub-pixel-weighted contour algorithm giving a series of points $(x, y)$ as a function of distance along the outline, from which curvature was calculated as $(x'y'' - y'x'')/(x'^2 + y'^2)^{3/2}$, where each partial derivative is with respect to distance along the outline. Code for contour tracing, curvature measurement and cluster intensity measurement are available at the Harvard Dataverse[48]. Curvatures $< -0.7$ or $> 0.7\,\mu m^{-1}$ were ignored. Cluster intensities were measured by refining a 2D Gaussian onto raw image data that had undergone local background subtraction with a 2.5-µm Gaussian filter. Unless otherwise indicated, curvature enrichment histograms were weighted with the reciprocal of the corresponding whole-data set curvature probability. Relative curvature enrichment was calculated by normalizing to the average cluster frequency between $-0.05$ and $0.05\,\mu m^{-1}$.

**Cryo-electron tomography.** Cells were applied to a hydrophilic lacey carbon film supported on a copper grid, blotted and plunge frozen into liquid ethane using a manual plunger[49]. TEM images were collected on a JEOL JEM3100FFC equipped with in-column Omega energy filter, FEG electron source, cryo-stage maintained at 86 K, using a Gating CCD camera at the end of a decelerator[50].

**Data availability.** The following data and analysis code have been deposited in the Harvard Dataverse: plasmid and protein sequences[47]; raw curvature and cluster intensity data[51]; and code for contour tracing, curvature measurement and cluster intensity measurement[48]. The authors declare that all other data supporting the findings of this study are available within the article and its Supplementary Information files, or from the corresponding author upon request.

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

## Acknowledgements

We thank John Sandy Parkinson (University of Utah, Salt Lake City) for helpful discussions and generous gifts of *E. coli* strains and plasmids, and thank Roseann Csencsits (University of California, Berkeley) and Kenneth Downing (University of California, Berkeley) for performing electron microscopy. We also thank John Sandy Parkinson and Kerwyn Casey Huang (Stanford University) for careful readings of the manuscript. Finally, we thank Quanming Shi (Stanford University) for discussions and extensive help with lithography and Rajarshi Ghosh (Stanford University) for many helpful discussions. This work was partially supported by the National Institutes of Health (NIH) Grant GM77856 (to J.L.) and NIH/National Cancer Institute (NCI) Grants U54CA143836 (to J.L.).

## Author contributions

W.D. and J.L. designed, performed and analysed experiments. W.D. and J.L. wrote the manuscript.

## Additional information

**Competing interests:** The authors declare no competing financial interests.

**Publisher's note**: 

