## [Peer Review File · Nature Communications]

Reviewers' comments:

Reviewer #1 (Remarks to the Author):

The chemotaxis system of *E. coli* is perhaps the most thoroughly investigated of all signaling systems. As such, it has been a productive focus of many quantitative studies. One of the earliest striking observations at the molecular level is that the chemotaxis receptors are clustered, with large clusters localized at both cell poles. This has the obvious advantage of ensuring each daughter cell inherits a functional receptor cluster. Moreover, a great deal of work has been devoted to characterizing and understanding receptor clustering, with great progress. However, the underlying biophysical mechanism for cluster localization has remained the subject of more speculation than science. The current work takes a big step in clarifying the biophysical mechanism for cluster localization. Specifically, the authors show that localization is determined by a preference of clusters for negative curvature (defined such that the poles have negative curvature) and that this preference increases with cluster size. The experiments productively exploit imaging of fluorescently tagged reporter in cells grown filamentously in confined chambers to induce large variations in cell curvature. The work is careful and rigorous, with many alternative scenarios - e.g. involving the Min-protein system, cardiolipin, or the Tol/Pal system - considered experimentally and excluded. By imaging many cells and quantifying localization separately for clusters of different sizes, the authors make the important observation that localization quantitatively obeys a model in which localization energy depends linearly on curvature, with a coefficient that increases with cluster size. The strong dependence on cluster size helps clarify an otherwise confusing set of previous mutant experiments in which localization was seen to vary - the current work makes it clear that in all these cases the change in localization can be attributed to decreases in cluster size, rather than changes in curvature preference. Based on these previous experiments and new mutant experiments, the authors argue that curvature preference arises from the entropic splay of transmembrane helices that are held together by protein-protein interactions in the cytoplasm. In what is in my view the strongest part of the work, the authors go on to test this proposal with synthetic constructs that implement this simple molecular architecture, and reproduce the observed localization pattern of wild-type chemoreceptors. Overall, the work clarifies a longstanding puzzle in an important model system, with general implications for localization of membrane-associated proteins. The work is generally of high quality and interest, and I am happy to recommend publication in *Nature Communications*.

I have one suggestion regarding the interpretation of data. In order to further test their proposed mechanism of curvature preference, the authors create a construct in which the transmembrane domains are held together in the periplasm rather than in the cytoplasm. The resulting clusters display a slight preference for positive curvature, but this is much weaker than the preference for negative curvature of the constructs held together in the cytoplasm. It seems likely that origin of this difference is that there is always negative curvature with respect to the cytoplasm associated with the circumferential direction of the cell, and this naturally constitutes a major difference between the energetics of the cytoplasmic and periplasmic constructs. A few remarks pointing out this difference seem to be in order.

Reviewer #2 (Remarks to the Author):

A- Summary of the key results.

In this work, the authors examine the sensitivity of chemoreceptor cluster localization to membrane curvature. They grow filamentous cells that present varying curvature degrees along the cell body, and observe a strong dependence of cluster localization on regions of positive curvature. The dependence is stronger as the size of the cluster increases. The authors propose that this dependence is due to the cone shape of the trimer of dimers, and consequently engineer

chemoreceptor-derived proteins and a completely unrelated protein. Then they analyze the localization of the engineered proteins in the presence or absence of a scaffold additional protein that forces the formation of differently shaped complexes. The observation of cone-shaped complexes as chemoreceptor trimers of dimers showing preference for positive curvature, and that of inverted cone shapes showing preferential localization in regions of negative curvature, leads them to conclude that indeed the curvature sensitivity is due to lateral steric pressure across the two bilayer leaflets, as had been proposed for other membrane-associated protein complexes.

B- Originality and interest.

The curvature of the membrane at the cell poles as driving force for localization of chemoreceptor clusters has been proposed some time ago and it represents one of the presently considered models to explain the polar localization together with the stochastic nucleation model. However, it is still lacking a sufficient amount of experimental evidence in support of one or the other model. In this work, the authors provide experimental evidence using a nice approach that allows them to observe the dependence of localization on membrane curvature. Moreover, they show that conditions previously considered to impair or reduce polar clustering still retain curvature dependence, albeit less obvious due to the drastic reduction in cluster size. Additionally, they show that altering the shape of chemoreceptor derived constructs by interaction with a scaffolding protein can change the preference from positive curvature to negative one, providing support to the hypothesis of the curvature as the main driving force.

Based on these considerations, I think that the manuscript present novel results and is of sufficiently broad interest to be published in Nature Communications.

Points C to F:

C- Data and methodology: validity of approach, quality of data, quality of presentation.

D- Appropriate use of statistics and treatment of uncertainties

E- Conclusions: robustness, validity, reliability

F- Suggested improvements: experiments, data for possible revision

The study shows itself as solid and complete. It is clearly written and the approach and results are clearly presented.

However, I have a couple of concerns whose clarification could help to provide better support to the proposed hypothesis.

In my opinion, the engineered constructs should be more exhaustively described.

a. Tsr Δ TSH3 is described as a "chemoreceptor Tsr where the trimerization domain was entirely replaced with the 11 amino acid SH3 recognition sequence". However, a close examination of the available sequence (pTac-Tsr-SH3.apc? Proper designation of the files in the manuscript would help the analysis) shows the replacement of 56 residues from the cytoplasmic domain of Tsr by 21 residues that provide the recognition site of the scaffold protein. The shown results about positive curvature preference depending on the co-expression of the scaffold protein are very clear. My concern has to do with how to ascertain that the construct has a rod shape, when the removed part of Tsr consists in the whole hairpin tip and there is no way to know whether the residues that follow the insertion in the C-helix of the cytoplasmic hairpin are able to retain their native interactions with the N-helix and the other dimer to form the 4-helix bundle. The removed region is, as I said, of 56 residues, but E391, which is the residue at the very tip of the native hairpin, is residue number 26 of the deletion, so that the deletion removes more residues from the N-helix than from the C-helix. I think this should be clarified, or at least discussed.

b. The construct Tsr Δ TPSH3 is described as a Tsr-derivative in which the cytoplasmic trimerization domain has been removed and the SH3 recognition sequence inserted into an exposed loop of the periplasmic domain. The examination of the available

sequence (pTac-Tsr-pSH3.ape?) shows the insertion of the same 21 residues in the periplasmic domain (with only one residue missing from the periplasmic domain of Tsr, so that it is a true insertion, not a replacement), but the trimerization domain is intact in that sequence. This issue should be clarified. If the trimerization domain was removed, then it would be nice, as in the previous case, to know whether the resulting construct still has a rod shape. If the trimerization domain was not removed, as I understand from the sequence, then the result is a bit more difficult to interpret. It is clear that in the presence of the scaffold protein in the periplasm the complexes formed show a preference to negative curvature. What is not so clear to me is whether these complexes do have the inverted cone shape that is schematically depicted in Fig. 5B, since the dimer-to-dimer interactions would be still in action.

In both types of constructs It maybe more useful, instead of removing the trimerization domain, to introduce mutations that disrupt trimer formation, and insert the SH3 recognition domain in the cytoplasmic domain in the position equivalent to E391.

Minor comments:

- line 39: the proper reference is not #15 but #28

- when referring to mutants derived from RP437, I think it is more correct to explicitly mention that they are mutants than to say "RP437 cells lacking such and such gene..."

- legend to Fig. 3B: clarify that the shown sample micrographs correspond to the delta cheA cheW cells, maybe including the words "sample micrographs" within the parenthesis that mentions the number of analyzed clusters. Were these cells grown in the presence of cephalixin? In figure 3C the non-filamentous character is explicitly mentioned, but I do not see a reason why not to grow the delta cheAW in the presence of cephalixin.

G- References. Appropriate credit to previous work?

- line 176: is #19 the proper reference? The glycine hinge in the cytoplasmic domain of chemoreceptors was first identified by Coleman et al, 2005

H- Clarity and context: lucidity of abstract/summary, appropriateness of abstract, introduction and conclusions

In summary, my opinion is that this work provides nice experimental support to the hypothesis of the membrane curvature as driving force for polar localization of chemoreceptor clusters. The findings are well exposed and emphasized in the abstract. Introduction and conclusions are appropriate.

Claudia Studdert

Reviewer #3 (Remarks to the Author):

This is an interesting biophysics/synthetic biology paper. Draper and Liphardt observed that bacterial chemoreceptor localization is highly sensitive to membrane curvature. Further, they proposed that this curvature sensitivity is a result of conformational entropy within chemoreceptor dimer-trimers spreading the TM regions apart, which in turn leads to the cone shape that favors curved bilayers. Furthermore, with this hypothetical entropic spreading model, the

authors developed artificial protein complexes built from un-curved components which can recognize either positive or negative membrane curvature, respectively.

The biological finding of that chemoreceptors preferentially localize at curved membranes is not novel. For example, Stahl et al. previously showed that the preference for strongly curved membranes arises from the curved shape of chemoreceptor trimer of dimers (Nat Commun. 2015 Nov 2;6:8728. doi: 10.1038/ncomms9728). Nonetheless, the entropic spreading model is quite interesting. Particularly, I was impressed by their designed synthetic complexes that employ this similar principle that can bind to either negative or positive curvature. I think that with more in depth mechanistic analyses, this paper should be potentially of interest to the wide readership of Nature Communications.

Some more specific comments and suggestions:

1. As the conformational entropy model is the key finding, I would suggest to include computational simulations (e.g. MD) with the trimers-of-dimers transmembrane domains in an explicit membrane environment. The results will likely not only shed molecular insight of the conformational changes and its impact on the surrounding phospholipid bilayers (i.e. membrane remodeling or curvature formation), but also provide further design guidelines for artificial curvature sensing proteins. Finally, it will also provide nice visualization of this paper and improve its readability.

2. To test the main conclusion of the manuscript, it would be helpful to introduce point mutations to the transmembrane domains as well as cytosolic region of the chemoreceptors. Previously, people applied cysteine and tryptophan scanning to various chemoreceptor to study their conformational change (For example, see "The PICM chemical scanning method for identifying domain-domain and protein-protein interfaces: applications to the core signaling complex of E. coli chemotaxis" Methods Enzymol. 2007;423:3-24). Would it be interesting to predict the roles of different point mutation based on the conformational entropy model and validate the prediction experimentally?

3. In general, the paper is well written. But this reviewer found some figures (e.g. Fig 2) a bit hard to interpret. This could be personal preference or due to different discipline-specific presentation styles. I might suggest to the authors to improve the presentation for Figure 2 and 3 for a general readership.

The chemotaxis system of E. coli is perhaps the most thoroughly investigated of all signaling systems. As such, it has been a productive focus of many quantitative studies. One of the earliest striking observations at the molecular level is that the chemotaxis receptors are clustered, with large clusters localized at both cell poles. This has the obvious advantage of ensuring each daughter cell inherits a functional receptor cluster. Moreover, a great deal of work has been devoted to characterizing and understanding receptor clustering, with great progress. However, the underlying biophysical mechanism for cluster localization has remained the subject of more speculation than science.

We agree! We believe that it is easy to forget how poorly understood cluster localization is, given how well studied the complex is in so many other respects.

The current work takes a big step in clarifying the biophysical mechanism for cluster localization. Specifically, the authors show that localization is determined by a preference of clusters for negative curvature (defined such that the poles have negative curvature) and that this preference increases with cluster size. The experiments productively exploit imaging of fluorescently tagged reporter in cells grown filamentously in confined chambers to induce large variations in cell curvature. The work is careful and rigorous, with many alternative scenarios - e.g. involving the Min-protein system, cardiolipin, or the Tol/Pal system - considered experimentally and excluded. By imaging many cells and quantifying localization separately for clusters of different sizes, the authors make the important observation that localization quantitatively obeys a model in which localization energy depends linearly on curvature, with a coefficient that increases with cluster size. The strong dependence on cluster size helps clarify an otherwise confusing set of previous mutant experiments in which localization was seen to vary - the current work makes it clear that in all these cases the change in localization can be attributed to decreases in cluster size, rather than changes in curvature preference.

Thanks - glad that you find the evidence for curvature driving sorting so convincing!

Based on these previous experiments and new mutant experiments, the authors argue that curvature preference arises from the entropic splay of transmembrane helices that are held together by protein-protein interactions in the cytoplasm. In what is in my view the strongest part of the work, the authors go on to test this proposal with synthetic constructs that implement this simple molecular architecture, and reproduce the observed localization pattern of wild-type chemoreceptors. Overall, the work clarifies a longstanding puzzle in an important model system, with general implications for localization of membrane-associated proteins. The work is generally of high quality and interest, and I am happy to recommend publication in Nature Communications.

Thank you for your enthusiasm for the work, and your close read of the manuscript.

I have one suggestion regarding the interpretation of data. In order to further test their proposed mechanism of curvature preference, the authors create a construct in which the transmembrane domains are held together in the periplasm rather than in the cytoplasm. The resulting clusters display a slight preference for positive curvature, but

this is much weaker than the preference for negative curvature of the constructs held together in the cytoplasm. It seems likely that origin of this difference is that there is always negative curvature with respect to the cytoplasm associated with the circumferential direction of the cell, and this naturally constitutes a major difference between the energetics of the cytoplasmic and periplasmic constructs. A few remarks pointing out this difference seem to be in order.

You bring up an excellent point that we didn't expand upon in the original manuscript. As you say, the 'periplasmic' complex is less curvature sensitive than the 'cytoplasmic' complex. A major factor is probably that the periplasmic complex has positive curvature with respect to a membrane that is always negatively curved, whereas the cytoplasmic complex has positive curvature with respect to a less positively curved membrane. The energetics of the periplasmic complex's curvature sign mismatch are likely much more complicated than the energetics of the cytoplasmic complex, in ways that current models don't attempt to account for. A second factor that we have previously considered is that the SH3 recognition sequence in the periplasmic complex is expected to be several-fold closer to the cell membrane than in the cytoplasmic complex. If the SH3 sequence acts like a hinge (which it should as it is bound to a flexible scaffold protein), we expect the periplasmic complex to experience less divergence at the membrane and thus less curvature enrichment. Additionally, the rod protein will experience less thermal deflection $\langle h \rangle$ along a shorter distance. We added a paragraph at line 230 to address these points.

Reviewer #2

A- Summary of the key results

In this work, the authors examine the sensitivity of chemoreceptor cluster localization to membrane curvature. They grow filamentous cells that present varying curvature degrees along the cell body, and observe a strong dependence of cluster localization on regions of positive curvature. The dependence is stronger as the size of the cluster increases. The authors propose that this dependence is due to the cone shape of the trimer of dimers, and consequently engineer chemoreceptor-derived proteins and a completely unrelated protein. Then they analyze the localization of the engineered proteins in the presence or absence of a scaffold additional protein that forces the formation of differently shaped complexes. The observation of cone-shaped complexes as chemoreceptor trimers of dimers showing preference for positive curvature, and that of inverted cone shapes showing preferential localization in regions of negative curvature, leads them to conclude that indeed the curvature sensitivity is due to lateral steric pressure across the two bilayer leaflets, as had been proposed for other membrane-associated protein complexes.

Thank you! To our knowledge, there are no proven examples where pressure-driven membrane curvature is mediated by an attractive force, rather, all known examples are driven by a repulsive steric force.

B- Originality and interest.

The curvature of the membrane at the cell poles as driving force for localization of chemoreceptor clusters has been proposed some time ago and it represents one of the presently considered models to explain the polar localization together with the stochastic nucleation model. However, it is still lacking a sufficient amount of

experimental evidence in support of one or the other model. In this work, the authors provide experimental evidence using a nice approach that allows them to observe the dependence of localization on membrane curvature. Moreover, they show that conditions previously considered to impair or reduce polar clustering still retain curvature dependence, albeit less obvious due to the drastic reduction in cluster size. Additionally, they show that altering the shape of chemoreceptor derived constructs by interaction with a scaffolding protein can change the preference from positive curvature to negative one, providing support to the hypothesis of the curvature as the main driving force. Based on these considerations, I think that the manuscript present novel results and is of sufficiently broad interest to be published in Nature Communications.

We appreciate your endorsement of the work, and your detailed reading of the manuscript.

Points C to F:

C- Data and methodology: validity of approach, quality of data, quality of presentation.

D- Appropriate use of statistics and treatment of uncertainties

E- Conclusions: robustness, validity, reliability

F- Suggested improvements: experiments, data for possible revision

The study shows itself as solid and complete. It is clearly written and the approach and results are clearly presented.

Thank you!

However, I have a couple of concerns whose clarification could help to provide better support to the proposed hypothesis.

In my opinion, the engineered constructs should be more exhaustively described.

a. Tsr Δ TSH3 is described as a “chemoreceptor Tsr where the trimerization domain was entirely replaced with the 11 amino acid SH3 recognition sequence”. However, a close examination of the available sequence (pTac-Tsr-SH3.ape? Proper designation of the files in the manuscript would help the analysis) shows the replacement of 56 residues from the cytoplasmic domain of Tsr by 21 residues that provide the recognition site of the scaffold protein. The shown results about positive curvature preference depending on the co-expression of the scaffold protein are very clear. My concern has to do with how to ascertain that the construct has a rod shape, when the removed part of Tsr consists in the whole hairpin tip and there is no way to know whether the residues that follow the insertion in the C-helix of the cytoplasmic hairpin are able to retain their native interactions with the N-helix and the other dimer to form the 4-helix bundle. The removed region is, as I said, of 56 residues, but E391, which is the residue at the very tip of the native hairpin, is residue number 26 of the deletion, so that the deletion removes more residues from the N-helix than from the C-helix. I think this should be clarified, or at least discussed.

Excellent points! To address the issues you raise, we have added significantly more detail about how each protein was constructed. First, we added a description.txt file to the sequence repository that describes the contents of each separate sequence file. We also cleaned up and clarified the annotations within the sequence files. Second, we added an extra supplementary figure, Fig. S5, with annotated protein sequences for all the components of the synthetic protein complexes. The figure shows an alignment for how each mutant “rod” Tsr relates to the wild-

type Tsr sequence, and additionally shows to the heterologous rod protein and scaffold proteins. Finally, we added extra detail to the maintext in the paragraphs beginning at line 191 and 213 explaining the construction.

We based our complete trimerization domain deletion on annotation from a bioinformatic analysis of heptad repeats in the coiled-coil region for all sequenced chemoreceptors (Alexander and Zhulin, 2007, PNAS 140, 2885). The researchers identified the signaling/trimerization domain as N and C helix heptads 1-4. In their annotation, they identified E391 as a pivot residue that does not belong in either helix, so we centered our deletion 25 amino acids on either side of E391, for a total of a 51 residue deletion. However, if we did delete an extra amino acid from the N helix, we expect that the coiled coil and 4-helical bundle would still form properly as we included two 5 amino acid flexible linkers on either side of the SH3 recognition sequence, which should allow some offset in the helices. While we fully expect such a deletion would have a dramatic impact on receptor signaling, we expect that the coiled coil domain would still form normally, as coiled coil domains are generally considered to be well behaved and very stable.

We expect that any chemoreceptor misfolding would expose hydrophobic residues from the 4-helical bundle core, which would lead to either protein degradation or aggregation. A primary motivation for doing cryo-TEM was to rule out aggregate formation, which should have been readily observable in our preparations. However, there was no evidence of aggregates even in gross overexpression. Additionally, we when expressed alone, the mutant Tsr Δ TSH3 (and Tsr Δ TpSH3) spread out evenly across the membrane, suggesting that the protein can freely diffuse within the membrane, and is thus un-aggregated. We added extra detail to the main text at line 201 to address the issue of proper rod protein folding.

b. The construct Tsr Δ TpSH3 is described as a Tsr-derivative in which the cytoplasmic trimerization domain has been removed and the SH3 recognition sequence inserted into an exposed loop of the periplasmic domain. The examination of the available sequence (pTac-Tsr-pSH3.ape?) shows the insertion of the same 21 residues in the periplasmic domain (with only one residue missing from the periplasmic domain of Tsr, so that it is a true insertion, not a replacement), but the trimerization domain is intact in that sequence. This issue should be clarified. If the trimerization domain was removed, then it would be nice, as in the previous case, to know whether the resulting construct still has a rod shape. If the trimerization domain was not removed, as I understand from the sequence, then the result is a bit more difficult to interpret. It is clear that in the presence of the scaffold protein in the periplasm the complexes formed show a preference to negative curvature. What is not so clear to me is whether these complexes do have the inverted cone shape that is schematically depicted in Fig. 5B, since the dimer-to-dimer interactions would be still in action.

Unfortunately there was a typographical error in the sequence file we uploaded, leading to confusion, for which we apologize. During annotation, the trimerization region of Tsr-pSH3 sequence was accidentally swapped for the native Tsr trimerization region. The correct Tsr-pSH3 sequence has the 56 amino acids centered on E391 replaced with a flexible linker, GGSGG. This mutation completely eliminates clustering, as shown in Fig. 5b, upper left. We have carefully combed through all the other sequence files to look for other errors, and no others were found.

In both types of constructs it may be more useful, instead of removing the trimerization domain, to introduce mutations that disrupt trimer formation, and insert the SH3

recognition domain in the cytoplasmic domain in the position equivalent to E391.

We did consider this approach, as we are aware of several point mutations that completely abolish trimer formation. However, we opted against such an approach over concerns that there could be residual, albeit dramatically reduced trimerization potential along the length of the trimerization domain in the case of single or even double mutants, as there are many intra-trimer contacts. The scaffold protein would hold the impaired trimerization domains in close proximity, such that this residual affinity could have an effect on complex formation. In other unrelated work we have done with scaffold proteins, we have had issues where a similar type of residual affinity between protein complexes with an estimated $K_d > 100 \mu\text{M}$ (!!) led to uncontrolled aggregation. To avoid even the possibility of these confounding issues, we adopted to remove the trimerization domain entirely. In the future, it could be very interesting for the broader community to build on these results, for example by pursuing a specific single-point mutation approach.

Minor comments:

- line 39: the proper reference is not #15 but #28

- when referring to mutants derived from RP437, I think it is more correct to explicitly mention that they are mutants than to say “RP437 cells lacking such and such gene...”

Yes! We fixed the reference and clarified our description of the mutants throughout the manuscript as per the reviewer’s suggestion.

- legend to Fig. 3B: clarify that the shown sample micrographs correspond to the delta cheA cheW cells, maybe including the words “sample micrographs” within the parenthesis that mentions the number of analyzed clusters. Were these cells grown in the presence of cephalixin? In figure 3C the non-filamentous character is explicitly mentioned, but I do not see a reason why not to grow the delta cheAW in the presence of cephalixin.

Indeed this was unclear in the original manuscript. To address this issue and clarity issues another reviewer raised, we rearranged the sample micrographs in Fig. 3 in a more systematic way. All of the micrographs in Fig. 3 are of cells grown in suspension without cephalixin. This is made confusing because Min knockout cells grow as short filaments, and thus appear cephalixin treated. We provide sample micrographs of cephalixin treated, curvy cells in Fig. S4.

G- References. Appropriate credit to previous work?

- line 176: is #19 the proper reference? The glycine hinge in the cytoplasmic domain of chemoreceptors was first identified by Coleman et al, 2005

Correct - the reference is now fixed!

Reviewer #3

This is an interesting biophysics/synthetic biology paper. Draper and Liphardt observed that bacterial chemoreceptor localization is highly sensitive to membrane curvature. Further, they proposed that this curvature sensitivity is a result of conformational entropy within chemoreceptor dimer-trimers spreading the TM regions apart, which in turn leads to the cone shape that favors curved bilayers. Furthermore, with this hypothetic entropic spreading model, the authors developed artificial protein complexes

built from un-curved components that can recognize either positive or negative membrane curvature, respectively.

The biological finding of that chemoreceptors preferentially localize at curved membranes is not novel. For example, Stahl et al. previously showed that the preference for strongly curved membranes arises from the curved shape of chemoreceptor trimer of dimers (Nat Commun. 2015 Nov 2;6:8728. doi: 10.1038/ncomms9728). Nonetheless, the entropic spreading model is quite interesting. Particularly, I was impressed by their designed synthetic complexes that employ this similar principle that can bind to either negative or positive curvature. I think that with more in depth mechanistic analyses, this paper should be potentially of interest to the wide readership of Nature Communications.

Thank you for taking the time read the work, and we appreciate your interest in the entropic spreading model!

While the observation of chemoreceptors in curved membranes is not novel, only a couple of publications to date raise the possibility of a causal mechanism, notably Strahl et al. (*Nat Commun. 2015 Nov 2;6:8728. doi: 10.1038/ncomms9728*) and a theoretical paper by Endres (*Biophys. J. 2009, 96, 453-463, doi: <http://dx.doi.org/10.1016/j.bpj.2008.10.021>*). In fact, considerable disagreement and uncertainty still exists within the field, highlighted by a recent publication from Santos et al. (*Mol. Microbiol. 2014, 92, 985-1004, doi:10.1111/mmi.12609*), who argue that the Tol/Pal complex is the sole determinant of chemoreceptor localization.

Some more specific comments and suggestions:

1. As the conformational entropy model is the key finding, I would suggest to include computational simulations (e.g. MD) with the trimers-of-timers transmembrane domains in an explicit membrane environment. The results will likely not only shed molecular insight of the conformational changes and its impact on the surrounding phospholipid bilayers (i.e. membrane remodeling or curvature formation), but also provide further design guidelines for artificial curvature sensing proteins. Finally, it will also provide nice visualization of this paper and improve its readability.

This is fundamentally an experimental study in which our goal is to measure and report the relationship between curvature and clustering (Figs. 2), experimentally rule out competing explanations (Figs. 3, 4), and use forward synthetic approaches and unrelated parts (Fig. 5, 6) to demonstrate what we believe to be a novel mechanism for curvature sorting in membranes.

Our hope is that these experimental data will motivate the extremely sophisticated computational simulations you suggest, such as the “*MD with the trimers-of-timers transmembrane domains in an explicit membrane environment*”. As you are aware, such a line of inquiry, done properly, is a major undertaking and constitutes a separate research program. We are only aware of two published molecular dynamic simulations of chemoreceptors in membranes, each of which was large undertaking and a publication in its own right. In its current form, the conformational entropy model we discuss is a simple non-quantitative heuristic, and judging from the experimental results, this simple heuristic is capturing important physical aspects of complex curvature sensing. We would be excited to collaborate with a lab in the future to expand upon the model as it stands to simulate complex behavior and to make quantitative theoretical predictions, but molecular dynamics goes beyond our capabilities and the general scope of this experimental study.

2. To test the main conclusion of the manuscript, it would be helpful to introduce point mutations to the transmembrane domains as well as cytosolic region of the chemoreceptors. Previously, people applied cysteine and tryptophan scanning to various chemoreceptor to study their conformational change (For example, see "The PICM chemical scanning method for identifying domain-domain and protein-protein interfaces: applications to the core signaling complex of *E. coli* chemotaxis" *Methods Enzymol.* 2007;423:3-24). Would it be interesting to predict the roles of different point mutation based on the conformational entropy model and validate the prediction experimentally?

We agree that a detailed structural study would be interesting, and have put thought and effort into such studies. Cysteine scanning allows pairs of regularly juxtaposed residues to be chemically crosslinked and measured on a gel, making it possible to determine their relative frequency of proximity. However, the best crosslinker for studying the native array geometry, TMEA, is highly selective for receptor trimers, and so would not work for our purposes, as we have eliminated trimers. Even when using an ideal crosslinker such as TMEA, the signal from crosslinking is likely boosted by the highly repetitive nature of chemoreceptor arrays (to the extent that arrays are visible in cryo-TEM.) We *could* use a more general crosslinker (such as diamide), but we worry that such signal would be quite difficult to interpret, especially given the non-repeating nature of the synthetic cluster. We think your original point is the decisive one—the main next step is for a molecular dynamics lab with suitable capabilities and computing horsepower for explicit/membrane calculations to perform such a calculation. The beauty of such a study would be that it would make very specific predictions about which bases to target for future crosslinking experiments, which would then set the stage for mutagenesis, labeling, etc.

3. In general, the paper is well written.

Thanks!

But this reviewer found some figures (e.g. Fig 2) a bit hard to interpret. This could be personal preference or due to different discipline-specific presentation styles. I might suggest to the authors to improve the presentation for Figure 2 and 3 for a general readership.

Yes—you are absolutely correct. We have revised the flow of Fig. 2 to clarify the progression of analysis, and also re-arranged Fig. 3, which another reviewer found confusing. We hope these versions are clearer.

REVIEWERS' COMMENTS:

Reviewer #2 (Remarks to the Author):

I have read the revised manuscript. I think that the authors have properly addressed the referees' concerns. In my opinion the revised manuscript is ready for publication.

Reviewer #3 (Remarks to the Author):

In general, this revised manuscript was improved. I think that this is a high quality, well written manuscript that should be published in Nature Communications.

I still think that the manuscript can benefit from additional computational and mutagenesis studies to further strengthen the mechanistic model as I suggested in the first round of review. The authors argued that they did not have capability for such computational studies (I actually think that this would be quite doable) and the cross linking results may be hard to interpret (I feel that even negative results may shed light on the proposed mechanism), which I do not fully agree. However, I do not think that these should prevent this manuscript from publishing.